

# Hypertrophic effects of low-load blood flow restriction training with different repetition schemes: a systematic review and meta-analysis

Victor S. de Queiros[1], Nicholas Rolnick[2,3], Brad J. Schoenfeld[2], Ingrid Martins de França[4], João Guilherme Vieira[5], Amanda Veiga Sardeli[6], Okan Kamis[7], Gabriel Rodrigues Neto[8,9], Breno Guilherme de Araújo Tinôco Cabral[1,10] and Paulo Moreira Silva Dantas[1,10]

[1] Graduate Program in Health Sciences, Federal University of Rio Grande do Norte (UFRN), Natal, Rio Grande do Norte, Brazil
[2] Department of Exercise Science and Recreation, CUNY Lehman College, New York, USA
[3] The Human Performance Mechanic, New York, New York, USA
[4] Graduate Program in Physiotherapy, Federal University of Rio Grande do Norte, Natal, Rio Grande do Norte, Brazil
[5] Graduate Program in Physical Education, Federal University of Juiz de Fora (UFJF), Juiz de Fora, Minas Gerais, Brazil
[6] Institute of Inflammation and Ageing, University of Birmingham, Birmingham, United Kingdom
[7] Faculty of Sports Sciences, Aksaray University, Aksaray, Türkiye
[8] Faculty Nova Esperança (FAMENE/FACENE), João Pessoa, Brazil
[9] University Center for Higher Education and Development, Campina Grande, Paraíba, Brazil
[10] Graduate Program in Physical Education, Federal University of Rio Grande do Norte, Natal, Rio Grande do Norte, Brazil

Corresponding author
Paulo Moreira Silva Dantas, pgdantas@icloud.com

## ABSTRACT

**Objective.** This systematic review and meta-analysis analyzed the effect of low-load resistance training (LL-RT) with blood flow restriction (BFR) versus high-load resistance training (HL-RT) on muscle hypertrophy focusing on the repetition scheme adopted.

**Methods.** Four databases were searched to identify randomized controlled trials that compared the effect of LL-RT with BFR versus HL-RT on muscle hypertrophy. Standardized mean differences (SMD) were pooled in a random effects meta-analysis.

**Results.** The overall analysis did not demonstrate significant differences between conditions (SMD = 0.046; $p = 0.14$). A similar result was observed when we separately analyzed studies that used sets to momentary muscle failure (SMD = 0.033; $p = 0.520$), sets of 15 repetitions (SMD = 0.005; $p = 0.937$) and a fixed repetition scheme composed of 75 repetitions (SMD = 0.088; $p = 0.177$). The analysis considering body region indicates no difference in lower limb exercise between HL-RT and LL-RT with BFR (SMD = 0.00066; $p = 0.795$) while upper limb exercise favors HL-RT (SMD = 0.231; $p = 0.005$).

**Conclusion.** LL-RT with BFR elicits muscle hypertrophy similar to HL-RT regardless of the employed repetition scheme, although there appears to be a small beneficial effect in favor of HL-RT in upper limb exercise.

## INTRODUCTION

High-load resistance training (HL-RT) programs (≥70% 1-repetition maximum (1-RM)) have been recommended to increase maximal strength development and muscle hypertrophy (*American College of Medicine and Sport, 2009*). However, low-load resistance training (LL-RT; <60% 1-RM) can be as effective as HL-RT for muscle hypertrophy (*Schoenfeld et al., 2017*). Muscle hypertrophy induced by LL-RT seems to be dependent on a substantial level of effort with sets conducted in close proximity of momentary muscle failure (*Weakley et al., 2023*). To illustrate, a recent study compared LL-RT and HL-RT (30% and 80% 1-RM, respectively) carried out to 60% of concentric failure or absolute concentric failure, equalizing the training volume. HL-RT and LL-RT to concentric failure conditions resulted in significant increases (7.7–8.1%) in quadriceps cross-sectional area after eight weeks of training, but there was no statistical change after LL-RT without concentric failure (*Lasevicius et al., 2022*).

The aforementioned results provide evidence that training in close proximity of momentary muscle failure may be more important in LL-RT compared to HL-RT for increasing muscular size. Considering the need to train close to momentary muscle failure to maximize hypertrophic responses during LL-RT, a relatively high number of repetitions is required. On the other hand, LL-RT with blood flow restriction (BFR) may elicit comparable muscle hypertrophy compared to HL-RT while requiring a lower number of repetitions (*e.g.*, 15 repetitions per set) (*Laurentino et al., 2012*; *Lixandrão et al., 2015*). For example, *Laurentino et al. (2012)* observed that LL-RT (20% 1-RM) with BFR using sets of 15 repetitions produced muscle hypertrophy similar to HL-RT (80% 1-RM) in a sample of untrained, healthy young participants. However, this is not a universal finding. *Lixandrão et al. (2015)* used the same number of repetitions per set, load, and occlusion pressure (80% arterial occlusion pressure (AOP)), but reported superior hypertrophy in the HL-RT (80% 1-RM) in young participants.

The divergence between results can be at least partly attributed to the limited sample size in some studies, potentially increasing the chance of a type II error. A previous meta-analysis did not identify statistical differences between HL-RT and LL-RT with BFR for muscle hypertrophy (*Lixandrão et al., 2018*). However, at the time of the search (January 2017), only 10 studies met the eligibility criteria established by the authors, limiting statistical power of the analyses. Since that time, a considerable number of studies have been published (*Biazon et al., 2019*; *Buckner et al., 2020*; *Centner et al., 2019*; *Centner et al., 2022*; *Cook et al., 2017*; *Cook et al., 2018*; *Cook & Cleary, 2019*; *Pereira et al., 2019*; *Jessee et al., 2018*; *Kataoka et al., 2022*; *May et al., 2022*; *Ramis et al., 2020*; *Reece et al., 2023*; *Teixeira et al., 2022*), making it possible to carry out analyses that achieve greater statistical power and thus draw stronger causal inferences. In addition, as excessive perceptual demands have been deemed a barrier to BFR exercise (*Rolnick et al., 2021*) and exercise to

momentary muscle failure induces greater discomfort and requires a greater effort than non-failure LL-RT with BFR (*de Queiros et al., 2023*), implementing repetition schemes that confer similar benefits as HL-RT without excessively elevating perceptual demands are of practical importance. Therefore, the aim of this systematic review and meta-analysis was to analyze the effect of LL-RT with BFR *versus* HL-RT on muscle hypertrophy. As a secondary objective, we sought to determine if the repetition scheme adopted or body region exercised during LL-RT with BFR influenced resultant hypertrophy as an array of repetition schemes have been used in the current body of literature in BFR protocols.

## METHODS

This systematic review was conducted in accordance with the Preferred Reporting Items for Systematic Review and Meta-analysis (PRISMA) recommendations (*Page et al., 2021*). We declare that an unreviewed version of this study is available as a preprint (https://www.researchsquare.com/article/rs-3419589/v1).

### Procedures
#### *Eligibility criteria*
Eligibility criteria were determined based on PICOS (population, intervention, comparator, outcomes, study design).

- Population: Healthy adults (≥18 years old), trained or untrained, of both sexes;
- Intervention: LL-RT (≤50% 1RM) with BFR programs with a minimum duration of six weeks.
- Comparator: HL-RT programs (≥70% 1RM), without BFR with a minimum duration of six weeks;
- Outcomes: Changes (pre- and post-training difference) in muscle size assessed *via* magnetic resonance imaging, computed tomography or ultrasound;
- Study type: Randomized clinical trials (within-subject and between subjects). We chose not to exclude within-subject studies from the analysis, since there seems to be no effect of cross-education on muscle hypertrophy (*Bell et al., 2023*). We included studies that did not report the randomization method. However, this aspect was considered when assessing the risk of bias (*Higgins et al., 2019*). We restricted studies to those written in English or Portuguese.

### Information sources and search strategy
The following databases were used to identify eligible studies for this review: Cochrane Library, EMBASE (Elsevier), MEDLINE (*via* PubMed®) and Web of Science Core Collection (Clarivate Analytics). The search strategy for each database is reported in Table S1. In addition, a comprehensive scan of the reference list of included studies (citation tracking) was performed to identify eligible studies (*Horsley, Dingwall & Sampson, 2011*). The searches were initially carried out on December 4, 2022 and updated on July 21, 2023.

### Selection process and data management
All studies identified in each database were exported and saved in a single file. Subsequently, the files were inserted into Rayyan® (http://rayyan.qcri.org), an open access software

designed to aid in the screening of titles and abstracts (*Ouzzani et al., 2016*). Subsequently, duplicates were identified and excluded using the software. After eliminating duplicates, two researchers (VSQ and OK) selected eligible studies based on title and abstract. Divergences in opinion were resolved by a third researcher (NR). The eligibility of the studies was then verified through a complete reading of the full texts. The references of eligible studies were reviewed to identify any additional relevant studies that were not identified in the previous steps to help ensure inclusion of all relevant studies.

## Data extraction

Two researchers (VSQ and NR) were responsible for data extraction. The following information was extracted and entered into an Excel spreadsheet (Microsoft Corporation, Redmond, WA, USA): sample size, characteristics of the participants, characteristics of the interventions (load (%1RM), volume (sets number and repetitions per set), frequency, duration, and pressures), method used to evaluate muscle hypertrophy, muscle group evaluated and main conclusions. In addition, the mean and standard deviation (SD) of baseline and post-intervention measurements of muscle hypertrophy were extracted. When standard error (SE) was reported, SE was converted to SD by the equation $SD = SE * (\sqrt{n})$. Five studies (*Ellefsen et al., 2015*; *Kataoka et al., 2022*); *Libardi et al., 2015*; *Teixeira et al., 2022*; *Yasuda et al., 2011*) presented the data graphically. Data from these studies were estimated using ImageJ software (*de Queiros et al., 2021*). In six studies (*Buckner et al., 2020*; *Centner et al., 2022*; *Cook et al., 2017*; *Pereira et al., 2019*; *Jessee et al., 2018*; *Kim et al., 2017*) the data either were not available or could not be extracted from graphs, and we therefore requested the data from the corresponding author *via* e-mail.

## Risk of bias

Two reviewers (VSQ and IMF) assessed the risk of bias using the Cochrane Risk of Bias Tool for Randomized Controlled Trials, Version 2 (RoB 2) (*Sterne et al., 2019*). RoB 2 assesses bias through five domains: (i) bias arising from the randomization process; (ii) prejudice due to deviations from intended interventions; (iii) bias due to lack of outcome data; (iv) bias in measuring the outcome; (v) bias in the selection of the reported result and more "general risk of bias". The risk of bias judgments for each domain are "low risk of bias", "some concerns", or "high risk of bias".

## Statistical analyses

The standardized mean difference (SMD) was adopted as an effect measure. The SMD was calculated from the mean difference ($\Delta$mean) reported in each intervention, divided by the pooled pre-intervention SD and multiplied by an adjustment for small samples calculated using the following formula: $[1 - (3/(4\times (n1 + n2 - 2) - 1)]$ (*Morris, 2008*). Due to methodological heterogeneity across studies, we employed a random effects model to estimate mean sizes and calculate 95% confidence intervals. The variance component for the random effects model was estimated from the restricted maximum likelihood (REML). As some individual studies had multiple effect sizes, we used robust variance estimation to account for dependent effect sizes (*Hedges, Tipton & Johnson, 2010*). Analyses were performed with the robumeta package (*Fisher et al., 2017*) implemented in R language.

We assumed a correlation of 0.80 between dependent effect sizes within the same study (*Fisher & Tipton, 2015*). A single study analyzed males and females separately (*Reece et al., 2023*); in this case, data were combined using the Review Manager calculator (Version 5.3). In addition, we used meta-regression to assess the variation in effect sizes based on repetition scheme used and muscle group evaluated across different studies. The statistical heterogeneity of treatment effects between studies was analyzed using the $I^2$ inconsistency test. Inconsistency was classified as: low (<25%), moderate (25–49%) and high (>50%) (*Higgins & Thompson, 2002*). Effects were considered statistically significant at a *p*-value of <0.05. ES between ≥0.2 and <0.5 were considered small, between ≥0.5 and <0.8 were considered medium, and ≥0.8 were considered large (*Sullivan & Feinn, 2012*).

### Certainty assessment

The quality of evidence was assessed using the Grading of Recommendations, Assessment, Development and Evaluation (GRADE) (*Zhang, Akl & Schünemann, 2019*). Initially, the quality of the evidence was classified as high (4 points). However, the quality of evidence was downgraded when: (i) the majority of studies included in the meta-analysis received a "some concerns" rating in RoB 2, (ii) when high and significant heterogeneity was identified in the meta-analysis or when there was minimal or no overlap of confidence intervals; (iii) when participants differed from the population of interest, when interventions differed from the specific intervention desired, or when surrogate outcomes were used instead of relevant outcomes; (iv) when a broad CI was identified that could impact the results.

## RESULTS

### Study selection

A total of 1,909 studies were identified in the databases. Twenty-three studies ultimately met inclusion criteria. The characteristics of excluded studies are presented in Table S2. Details of the screening process are reported in Fig. 1.

### Subjects

Four hundred and ninety-five individuals were included in the analysis. Five studies evaluated older individuals (average age > 60 years) (*Cook et al., 2017*; *Cook & Cleary, 2019*; *Pereira et al., 2019*; *Libardi et al., 2015*; *Vechin et al., 2015*), while the other studies evaluated young individuals. Twelve studies included only men (*Biazon et al., 2019*; *Centner et al., 2019*; *Centner et al., 2022*; *Kim et al., 2017*; *Kubo et al., 2006*; *Laurentino et al., 2022*; *Lixandrão et al., 2015*; *May et al., 2022*; *Ozaki et al., 2013*; *Ramis et al., 2020*; *Teixeira et al., 2022*; *Yasuda et al., 2011*), two studies included only women (*Ellefsen et al., 2015*; *Pereira et al., 2019*) and eight studies included mixed samples (*Buckner et al., 2020*; *Cook et al., 2017*; *Cook et al., 2018*; *Cook & Cleary, 2019*; *Jessee et al., 2018*; *Kataoka et al., 2022*; *Libardi et al., 2015*; *Vechin et al., 2015*) and one study analyzed men and women separately (*Reece et al., 2023*). Only one study investigated samples composed of trained individuals (*Kataoka et al., 2022*).
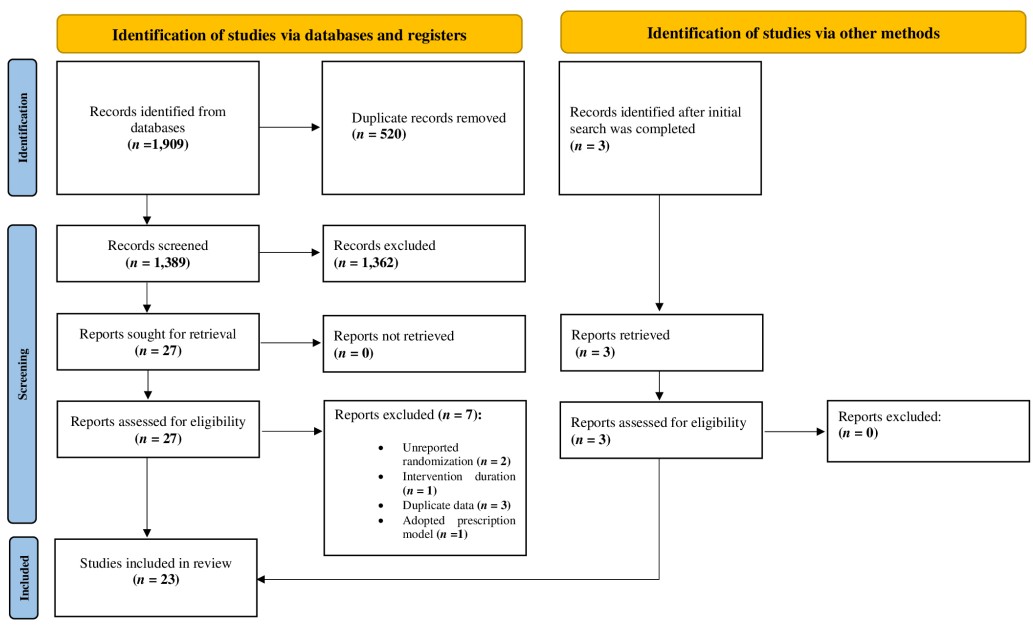

**Figure 1  PRISMA flow diagram.** Summary of the systematic literature search and study selection process.

## Study characteristics

The average duration of training interventions was 9.6 weeks, ranging from 6 (*Cook et al., 2018*; *Kataoka et al., 2022*; *Ozaki et al., 2013*; *Reece et al., 2023*; *Yasuda et al., 2011*) to 16 weeks (*Pereira et al., 2019*). All studies adopted a relatively low training frequency (2-3 times per week) in both conditions tested (LL-RT with BFR and HL-RT). Five studies evaluated upper limb muscles (*Buckner et al., 2020*; *Kim et al., 2017*; *Ozaki et al., 2013*; *Ramis et al., 2020*; *Yasuda et al., 2011*). For LL-RT with BFR, most studies adopted intensities (%1RM) recommended for LL-RT with BFR (20–40% 1RM) (*Patterson et al., 2019*), however two studies adopted intensities lower than recommended (15% 1RM) (*Buckner et al., 2020*; *Jessee et al., 2018*). For HL-RT, most studies (22/23) adopted the intensity (70–85% 1-RM) recommended by the (*American College of Medicine and Sport, 2009*) for muscle hypertrophy in novice and intermediate individuals. Eight studies adopted a protocol using sets carried out to momentary muscle failure in LL-RT with BFR and HL-RT (*Buckner et al., 2020*; *Cook et al., 2017*; *Cook et al., 2018*; *Cook & Cleary, 2019*; *Ellefsen et al., 2015*; *Jessee et al., 2018*; *Kataoka et al., 2022*; *Reece et al., 2023*). One study adopted a volume of four sets of eight repetitions in the HL-RT (80% 1-RM), while LL-RT with BFR included enough repetitions to equalize the total training volume between conditions (*Ramis et al., 2020*). The other studies adopted pre-defined repetition schemes in both conditions. Most studies individualized restriction pressures based on arterial occlusion pressure (AOP; 40–80% AOP), with three studies analyzing the impact of different pressures (*Buckner et al., 2020*; *Jessee et al., 2018*; *Lixandrão et al., 2015*). Arbitrary pressures were used in five studies (*Cook et al., 2018*; *Ellefsen et al., 2015*; *Kubo et al., 2006*; *Ozaki et al., 2013*;

*Yasuda et al., 2011*). None of the studies adopted BFR induced by non-pneumatic elastic bands/straps (*e.g.*, "practical BFR"). Two studies used intermittent BFR (*e.g.*, pressure released during rest intervals) (*Biazon et al., 2019*; *Pereira et al., 2019*). The characteristics of the studies are reported in Table 1.

## Quantitative analyses (meta-analysis)

Twenty-three studies and sixty-four effect sizes were included in the main meta-analysis, excluding *Kim et al. (2017)* as we had no access to the data of this study (mean pre-intervention and post-intervention and SD pre-intervention). This analysis indicated no difference between the tested conditions (LL-RT with BFR *versus* HL-RT) for muscle hypertrophy (SMD = 0.046 (95% CI [−0.015–0.1]); $p = 0.14$; $I^2 = 0\%\%$; Fig. 2). The results remained consistent when the studies were isolated considering the repetition schemes adopted in the LL-RT with BFR, that is momentary muscle failure (SMD = 0.033 (95% CI [−0.084–0.152]); $p = 0.520$; $I^2 = 0\%$), multiple sets of 15 repetitions per set (SMD = 0.005 (95% CI [−0.195–0.206]); $p = 937$; $I^2 = 0\%$)) and 75 repetitions segmented into four sets (*e.g.*, 30-15-15-15) (SMD = 0.088 (95% CI [−0.053–0.229]); $p = 0.177$; $I^2 = 0\%$). When studies were divided based on muscle group (upper limb *versus* lower limb), no differences were reported in analyses involving lower limb muscles (SMD = 0.00066 (95% CI [−0.046–0.059]); $p = 0.795$; $I^2 = 0\%$). On the other hand, analyses involving studies that evaluated upper limb muscles identified statistical differences favoring HL-RT (SMD = 0.231 (95% CI [0.138–0.324]); $p = 0.005$; $I^2 = 0\%$). The results of the subgroup analyses are reported in Table 2. As mentioned above, we were unable to include the data from *Kim et al. (2017)* in our upper limb subgroup analysis, as the corresponding author did not respond to our request for data.

## Risk of bias and quality of the evidence

The risk of bias assessment of each study is reported in Fig. 3. The risk of bias (overall) was classified as high in sixteen studies. The classification "some concerns" was assigned in seven studies. None of the included studies were classified as having a low risk of bias. Only two studies reported details about the randomization process (*Centner et al., 2022*; *Pereira et al., 2019*). Only three studies reported prospective trial registration (*Centner et al., 2022*; *Pereira et al., 2019*; *Reece et al., 2023*). Blinding of outcome assessors was performed in nine studies (*Buckner et al., 2020*; *Centner et al., 2019*; *Centner et al., 2022*; *Cook et al., 2017*; *Cook & Cleary, 2019*; *Ellefsen et al., 2015*; *Jessee et al., 2018*; *Kataoka et al., 2022*; *Ramis et al., 2020*). Considering the risk of bias present in the studies included in our meta-analyses, for the main meta-analysis, the quality of the evidence was moderate.

## DISCUSSION

The aim of this systematic review and meta-analysis was to analyze the effect of LL-RT with BFR *versus* HL-RT on muscle hypertrophy. As a main result, we found that muscle hypertrophy is similar between LL-RT with BFR *versus* HL-RT and is consistent with the results presented in a previously published meta-analysis on the topic (*Lixandrão et al., 2018*). The previous meta-analysis encompassed 10 studies, while our main meta-analysis

**Table 1   Summary of studies investigating muscle hypertrophy between high-load resistance training *vs.* low-load resistance training with blood flow restriction.**

| Reference | Sample | Study design | Duration (Frequency) | Exercise (s) | Exercise load (%1RM) | Volume (Inter-set rest [s]) | Pressure (cuff width) | Muscle mass assessment | Muscle group |
|---|---|---|---|---|---|---|---|---|---|
| *Biazon et al. (2019)* | Untrained young men (n = 30) | Within/between subjects | 10 weeks (2x) | Unilateral knee extension | LL+BFR: 20% HL:80% | LL+BFR: 3–4 x 20 (60) HL: 3–4 x 10 (60) | 60% AOP (17.5 cm) | Ultrasound | Vastus lateralis |
| *Buckner et al. (2020)* | Untrained young adults (n = 40) | Within/between subjects | 8 weeks (2x) | Unilateral elbow flexion | LL+BFR: 15% HL:70% | LL+BFR: 4 x failure (30) HL: 4 x failure (90) | 40, 80% AOP (5 cm) | Ultrasound | Arm |
| *Centner et al. (2022)* | Untrained young men (n = 29) | Between subjects | 14 weeks (2x) | Bilateral knee extension and leg press | LL+BFR: 20–35% HL:70–85% | LL+BFR: 1 x 30 + 3 x 15 (60) HL: 6–12 (60) | 50% AOP (12 cm) | MRI | Rectus femoris |
| *Centner et al. (2019)* | Untrained young men (n = 25) | Between subjects | 14 weeks (3x) | Plantar flexion | LL+BFR: 20–35% HL:70–85% | LL+BFR: 1 x 30 + 3 x 15 (60) HL: 6–12 (60) | 50% AOP (12 cm) | Ultrasound | Gastrocnemius |
| *Cook & Cleary (2019)* | Untrained older adults (n = 21) | Between subjects | 12 weeks (2x) | Bilateral knee extension and knee flexion | LL+BFR: 30% HL:70% | LL+BFR: 1–3 x failure (60) HL: 1–3 x failure (60) | 1.5 x SBP (6 cm) | MRI | Quadriceps Hamstrings |
| *Cook et al. (2017)* | Untrained older adults (n = 24) | Between subjects | 12 weeks (2x) | Bilateral knee extension, knee flexion, leg press horizontal | LL+BFR: 30–50% HL:70% | LL+BFR: 1–3 x failure (60) HL: 1–3 x failure (60) | 1.5 x SBP (6 cm) | MRI | Quadriceps |
| *Cook et al. (2018)* | Untrained young adults (n = 12) | Between subjects | 6 weeks (3x) | Bilateral knee extension and leg press horizontal | LL+BFR: 20% HL:70% | LL+BFR: 2 x 25 + 1 x failure (30) HL: 2 x 10 + 1 x failure (30) | 180-200 mmHg (5.4 cm) | MRI | Quadriceps |
| *Ellefsen et al. (2015)* | Untrained young women (n = 15) | Within-subject | 12 weeks (2x) | Unilateral knee extension | LL+BFR: 30% HL: 70–92% | LL+BFR: 5 x Failure (45) HL: 3 x 6–10 (90) | 90–100 mmHg (18 cm) | MRI | Quadriceps |
| *Jessee et al. (2018)* | Untrained young adults (n = 40) | Within/between subjects | 8 weeks (2x) | Unilateral knee extension | LL+BFR: 15% HL:70% | LL+BFR: 4 x failure (30) HL: 4 x failure (90) | 40, 80% AOP (10 cm) | Ultrasound | Thigh |
| *Kataoka et al. (2022)* | Trained young adults (n = 27) | Within-subject | 6 weeks (3x) | Plantar flexion | LL+BFR: 30% HL:70% | LL+BFR: 4 x failure (30) HL: 4 x failure (60) | 40% AOP (12 cm) | Ultrasound | Gastrocnemius |
| *Kim et al. (2017)* | Untrained young men (n = 9) | Within-subjects | 8 weeks (3x) | Unilateral elbow flexion | LL+BFR: 30% HL:75% | LL+BFR: 1 x 30 + 3 x 15 (30) HL: 3 x 10 (60) | 50% AOP (5 cm) | Ultrasound | Arm |
| *Kubo et al. (2006)* | Young men (n = 9) | Within-subjects | 12 weeks (3x) | Unilateral knee extension | LL+BFR: 20% HL:80% | LL+BFR: 25 − 18 − 15 − 12 (30) HL: 4 x 10 (30) | 180–250 mmHg NR | MRI | Quadriceps |
| *Laurentino et al. (2022)* | Untrained young men (n = 19) | Between subjects | 8 weeks (2x) | Bilateral knee extension | LL+BFR: 20% HL:80% | LL+BFR: 4 x 15 (60) HL: 4 x 8 (60) | 80% AOP (17.5 cm) | MRI | Quadriceps |
| *Libardi et al. (2015)* | Untrained older adults (n = 18) | Between subjects | 12 weeks (2x) | Leg Press | LL+BFR: 20–30% HL:70–80% | LL+BFR: 1 x 30 + 3 x 15 (60) HL: 4 x 8 (60) | 50% AOP (17.5 cm) | MRI | Quadriceps |
| *Lixandrão et al. (2015)* | Untrained young men (n = 26) | Within/between subject | 12 weeks (2x) | Unilateral knee extension | LL+BFR: 20, 40% HL: 80% | LL+BFR: 2–3 x 15 (60) HL: 2–3 x 10 (60) | 40, 80% AOP (17.5 cm) | MRI | Quadriceps |
| *May et al. (2022)* | Untrained young men (n = 17) | Between subjects | 7 weeks (3x) | Bilateral knee extension and knee flexion | LL+BFR: 20% HL:70% | LL+BFR: 1 x 30 + 3 x 15 (30) HL: 4 x 8 (120) | 60% AOP (10 cm) | pQCT | Quadriceps Hamstrings |
| *Ozaki et al. (2013)* | Untrained young men (n = 19) | Between subjects | 6 weeks (3x) | Bench press | LL+BFR: 30% HL:75% | LL+BFR: 1 x 30 + 3 x 15 (30) HL: 3 x 10 (120–180) | 100-160 mmHg (NR) | MRI | Triceps brachii Pectoralis major |
| *Pereira et al. (2019)* | Untrained older women (n = 18) | Between subjects | 16 weeks (2x) | Squat | LL+BFR: 30% HL:70% | LL+BFR: 4 x 15 (30) HL: 3 x 10 (60) | 50% AOP (18 cm) | pQCT | Quadriceps |
| *Ramis et al. (2020)* | Untrained young men (n = 28) | Between subjects | 8 weeks (3x) | Unilateral elbow flexion and knee extension | LL+BFR: 30% HL:80% | LL+BFR: 4 x 20.7 (120) HL: 4 x 8 (120) | 100% SBP + 20 mmHg (14 cm) 100% AOP + 40 mmHg (16 cm) | Ultrasound | Biceps brachii Quadriceps |
| *Reece et al. (2023)* | Recreationally active young adults (n = 27) | Between subjects | 6 weeks (3x) | Knee extension | LL+BFR: 30% HL:80% | LL+BFR: 3 x Failure (60) HL: 3 x 8–12 RM (120) | 50% AOP (10 cm) | Ultrasound | Vastus lateralis |
| *Teixeira et al. (2022)* | Physically active young men (n = 16) | Within-subject | 8 weeks (2x) | Unilateral knee extension | LL+BFR: 20% HL:70% | LL+BFR: 3 x 15 (60) HL: 3 x 8 (60) | 80% AOP (17.5 cm) | MRI | Quadriceps |
| *Vechin et al. (2015)* | Untrained older adults (n = 16) | Between subjects | 12 weeks (2x) | Leg press 45° | LL+BFR: 20–30% HL:70–80% | LL+BFR: 1x30 + 3x15 (60) HL: 4 x 8 (60) | 50% AOP (18 cm) | MRI | Quadriceps |
| *Yasuda et al. (2011)* | Untrained young men (n = 20) | Between subjects | 6 weeks (3x) | Bench press | LL+BFR: 30% HL:75% | LL+BFR: 1 x 30 + 3 x 15 (30) HL: 3 x 10 (120–180) | 100–160 mmHg (NR) | MRI | Triceps brachii Pectoralis major |

**Notes.**

*1RM, One-repetition maximum dynamic strength; AOP, Arterial occlusion pressure; HL, High-load; LL+BFR, Low-load with blood flow restriction; MRI, Magnetic resonance imaging; NR, Not reported; pQCT, Peripheral quantitative computed tomography; SBP, systolic blood pressure.

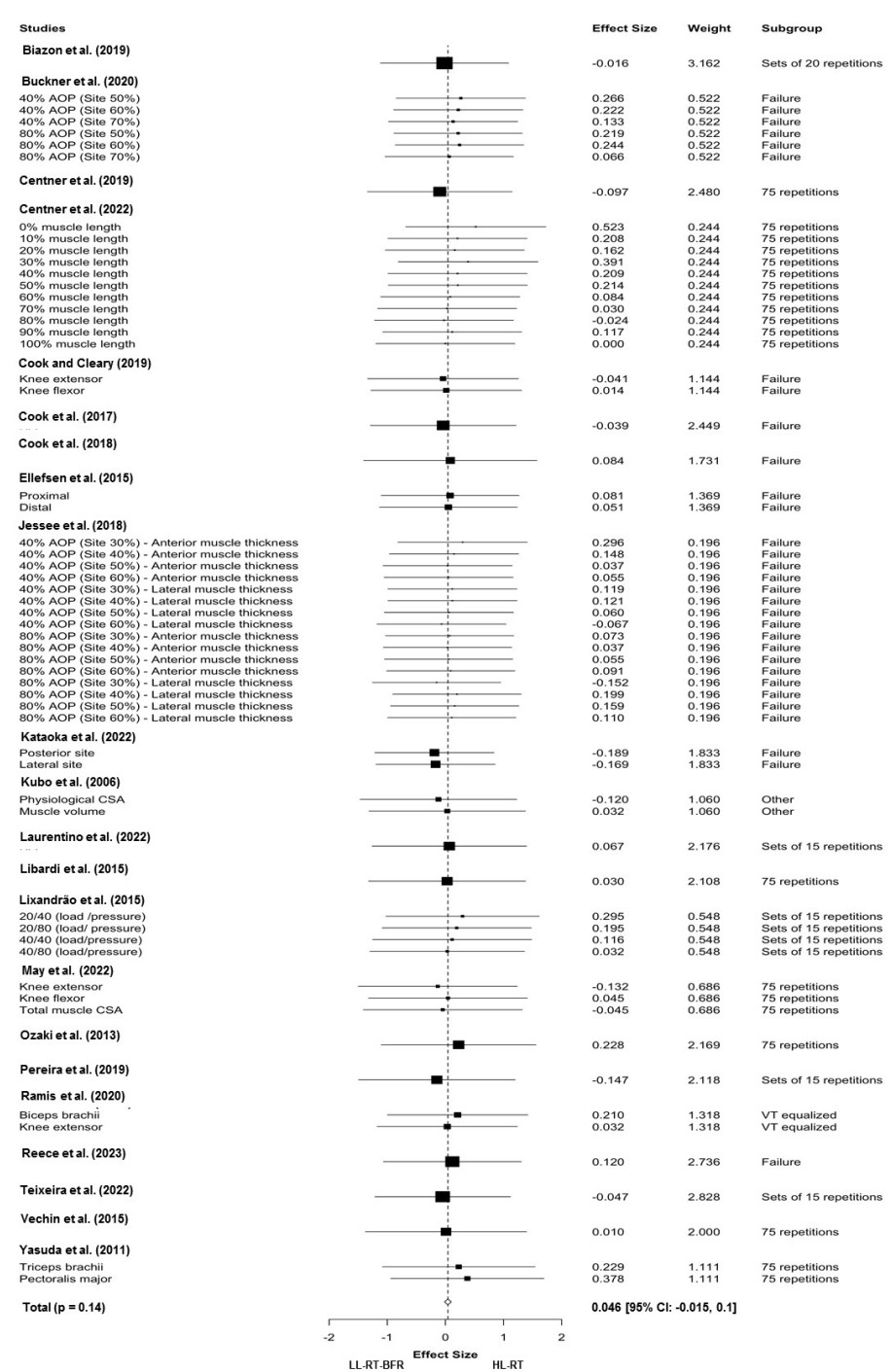

**Figure 2** **Forest plot demonstrating the effects of LL-RT with BFR *versus* HL-RT on muscle hypertrophy.** AOP, arterial occlusion pressure; HL-RT, high-load resistance training; LL-RT-BFR, low-load resistance training with blood flow restriction.

**Table 2  Random-effects meta-regression analysis of the prescribed repetition scheme and muscle group.**

|  | SMD | 95% CI | *p*-value |
|---|---|---|---|
| **Repetition scheme** | | | |
| 15 repetitions | 0.005 | −0.195, 0.206 | 0.937 |
| 75 repetitions | 0.088 | −0.053, 0.229 | 0.177 |
| Failure | 0.033 | −0.084, 0.152 | 0.520 |
| **Muscle group** | | | |
| Upper limb | 0.231 | 0.138, 0324 | 0.005 |
| Lower limb | 0.00066 | −0.046, 0.059 | 0.795 |

**Notes.**
CI, confidence interval.

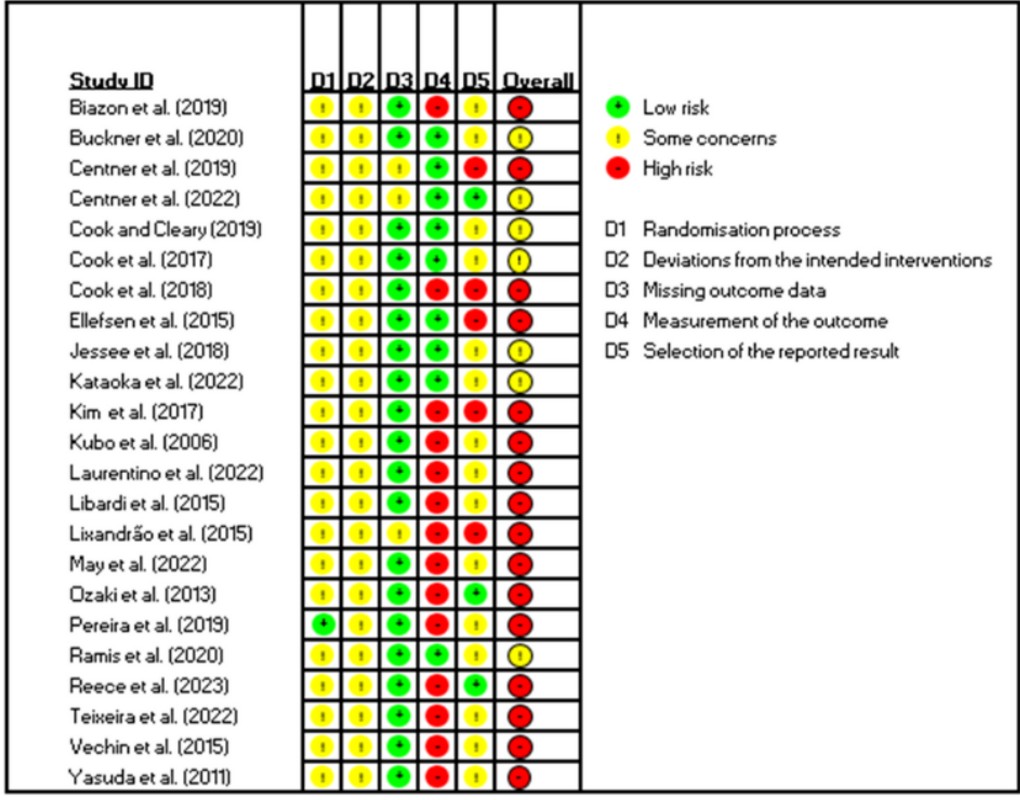

**Figure 3  Graph of risk of bias for the studies included in the in the review. .**

encompassed 22 studies. This large number of studies strengthens our ability to draw inferences on the effect of LL-RT with BFR *versus* HL-RT on muscle hypertrophy using a variety of repetition schemes. It is noteworthy that the effect of LL-RT with BFR *versus* HL-RT on muscle hypertrophy seems to be unaffected by the repetition scheme adopted in LL-RT with BFR, as shown in our subgroup analyses. However, we did observe a small beneficial effect in favor of HL-RT when using upper limb BFR exercise compared to lower limb BFR exercise.

HL-RT has been recommended for enhancement of muscle strength and size (*American College of Medicine and Sport, 2009*). Our overall analysis indicates that LL-RT with BFR elicits a similar magnitude of muscle hypertrophy compared to HL-RT. Mean percentage gains were similar between the compared training models (4.12% *versus* 5.8% for LL-RT with BFR and HL-RT, respectively). These results suggest that LL-RT with BFR may be a viable alternative to HL-RT to elicit muscle hypertrophy in healthy populations ranging from young to older adults. While the mechanisms underpinning the hypertrophic effects of LL-RT with BFR are still not fully elucidated, there appears to be some relationship with acute muscular fatigue and long-term muscle hypertrophy. Considering that LL-RT with BFR restricts arterial inflow and occludes venous return, exercise with BFR prevents the escape of metabolites from the working muscles, inducing earlier fatigue and thus conceivably increasing recruitment of motor units to maintain muscle strength levels (*Loenneke et al., 2011*) . This mechanism is theorized to be responsible for the higher myoelectric activity in low-load resistance exercise with BFR compared to low-load resistance exercise without BFR (work-matched) (*Takarada et al., 2000*). On the other hand, a recent meta-analysis found that LL-RT with BFR elicits lower myoelectric activity than HL-RT (*Cerqueira et al., 2022*).

Lower myoelectric activity in LL-RT with BFR can be justified by motor unit cycling referring to the fact that during exercise with lower loads, some motor units are activated and deactivated to minimize fatigue, reducing the need for all motor units to be activated at the same time (*Dankel et al., 2017*). It should be noted that surface electromyography amplitudes do not exclusively reflect the recruitment of motor units during exercise (*Vigotsky et al., 2018*); thus, inferences regarding recruitment must be drawn with caution.

It has been hypothesized that training to concentric failure elicits greater recruitment of motor units and, consequently, enhances muscle hypertrophy (*Fisher et al., 2011*). However, this claim may be load-dependent. Training close to concentric muscle failure may be necessary to maximize muscle hypertrophy in LL-RT, but not necessarily in HL-RT (*Lasevicius et al., 2022*). *Lixandrão et al. (2015)* found that LL-RT with BFR elicited significantly less quadriceps femoris hypertrophy than HL-RT, using low and high BFR pressures (40% and 80% AOP, respectively) and an arbitrary repetition scheme. On the other hand, *Jessee et al. (2018)* found similar hypertrophy of the quadriceps femoris between HL-RT using the same BFR pressures but with sets carried out to momentary muscle failure. *Jessee et al. (2018)* proposed that such a discrepancy could be explained by the repetition schemes prescribed in each intervention (three sets of 15 repetitions *versus* sets to momentary muscle failure).

Based on the premise that the repetition scheme adopted in the intervention could potentially influence muscle hypertrophy elicited by LL-RT with BFR, we chose to stratify our analyses based on the repetition scheme adopted in LL-RT with BFR. We categorized repetition schemes into three subgroups, including two that are recommended for the prescription of LL-RT with BFR; that is, sets carried out to momentary muscle failure and a fixed repetition scheme composed of 75 repetitions performed across four sets (30-15-15-15) (*Patterson et al., 2019*). The third repetition scheme analyzed was composed of multiple sets of 15 repetitions. In all subgroups investigating repetition schemes, no

statistical differences in muscle hypertrophy were observed between LL-RT with BFR and HL-RT.

In sets carried out to momentary muscle failure, LL-RT with BFR elicited a mean percentage increase of 3.84%, while HL-RT elicited a mean percentage increase of 5.3%. Similar results were observed in the analysis of the studies that employed a protocol of 75 repetitions in LL-RT with BFR (4.2% and 5.7% for LL-RT with BFR and HL-RT, respectively), as well as in the analysis that employed sets of 15 repetitions in LL-RT with BFR (4.9% and 5.8% for LL-RT with BFR and HL-RT, respectively). These results suggest that the muscle hypertrophy elicited by LL-RT with BFR is not necessarily dependent of the repetition scheme. In support of this hypothesis, *Martín-Hernández et al. (2013)* reported that the prescription of 75 or 150 repetitions (twice the repetition volume traditionally prescribed in practice) in LL-RT with BFR promotes muscle hypertrophy similar to HL-RT, with no difference between the different volumes of repetitions tested in BFR conditions.

It has been speculated that after the muscle reaches a certain level of fatigue during exercise with BFR, increasing number of repetitions is not of great relevance for muscle hypertrophy, suggesting the existence of a ceiling effect (*Counts et al., 2016*). Possibly, a considerable level of fatigue can be experienced with multiple sets of 15 repetitions and loads of 20–40% of 1RM with applied BFR pressures of 40–80% of AOP.

In addition, to identify potential region-specific changes in muscle growth between HL-RT and LL-RT, we introduced a subgroup analysis that considered the muscle group evaluated (upper limb and lower limb). Interestingly, we found a small, but statistically significant difference favoring HL-RT in the upper limb muscles while no differences between conditions were observed in muscle hypertrophy in the lower limbs. It is important to highlight that there is a paucity of studies comparing the effects of HL-RT and LL-RT with BFR on upper body hypertrophy ($n = 4$; 10 comparisons) in our meta-analysis.

Moreover, only one of these studies adopted a personalized pressure (%AOP) (*Buckner et al., 2020*) yet used a load below what is recommended (15% 1-RM) to induce similar muscle growth as HL-RT (*Patterson et al., 2019*). In addition, two studies used arbitrary pressures (100–160 mmHg). We speculate that the limited number of comparisons in the upper limb subgroup analysis may have contributed to the small beneficial effect observed in favor of HL-RT and the results would likely be attenuated if the number of comparisons were similar to lower limb subgroup ($n = 53$; 19 studies). More research comparing the hypertrophic response between LL-RT with BFR and HL-RT during upper limb exercise is needed given our findings.

This review has some notable strengths. A relatively large number of studies were included ($n = 22$) in the principal meta-analysis and a low-level of inconsistency was reported in all analyses performed. It is worth adding that all studies included in this review investigated local muscle growth through images obtained by ultrasound, magnetic resonance imaging, and peripheral computed tomography, improving the sensitivity to identify subtle changes in muscle mass (*Tavoian et al., 2019*).

Nevertheless, the present study has some limitations that need to be highlighted: (i) most of the measurements were taken at a single site along the length of the muscle, which may not reflect hypertrophic changes throughout the entire muscle; (ii) none of the

included studies had a low risk of bias; (iii) only one of the studies included in our analyses included trained individuals (*Kataoka et al., 2022*) and the interventions ranged from 6 to 16 weeks; therefore, our results cannot be extrapolated to trained individuals or long-term adaptations.

## PRACTICAL APPLICATIONS

As increased proximity to failure heightens the perceptual experiences of the exerciser regardless of the application of BFR (*Refalo et al., 2023*), it can be assumed that long-term adherence to repetition schemes further away from failure would be greater than those repetition schemes exercising with a greater volume of repetitions or to momentary muscular failure, although this requires further research. If hypertrophy is similar between different LL-RT with BFR repetition schemes compared to HL-RT, this may have important implications for injured individuals rehabilitating from injuries whose tolerance to strenuous exercise is reduced and LL-RT with BFR is recommended.

Tolerance to LL-RT with BFR has been labeled as a barrier to long-term compliance to the intervention (*Rolnick et al., 2021*). Therefore, reducing the required number of repetitions needed to induce positive musculoskeletal benefit seems important. *Patterson et al. (2019)* recommends sets either be carried out to momentary muscular failure or performed for 75 repetitions over 4 sets. The current systematic review with meta-analysis indicates that a smaller repetition volume (*e.g.*, multiple sets of 15) may induce similar hypertrophic effects as HL-RT.

## OTHER INFORMATION

### Registration and protocol

The original protocol was prospectively registered (CRD42022375960) in the International Prospective Register of Systematic Reviews (PROSPERO).

### Funding

The Improvement of Higher Education Personnel (CAPES) and the Research Support Foundation of the State of Minas Gerais (FAPEMIG) supported our study. The funders had no role in study design, data collection and analysis, decision to publish, or preparation of the manuscript.

### Grant Disclosures

The following grant information was disclosed by the authors:
Improvement of Higher Education Personnel (CAPES).
Research Support Foundation of the State of Minas Gerais (FAPEMIG).

### Competing Interests

Nicholas Rolnick is the founder of THE BFR PROS, a BFR education company that provides BFR training workshops to fitness and rehabilitation professionals across the

world using a variety of BFR devices. Nicholas Rolnick has no financial relationships with any cuff manufacturers/distributors. Brad J. Schoenfeld is on the scientific advisory board for Tonal, a manufacturer of fitness equipment. The other authors declare no potential or actual conflicts of interest.

## Author Contributions

- Victor S. de Queiros conceived and designed the experiments, performed the experiments, analyzed the data, prepared figures and/or tables, authored or reviewed drafts of the article, and approved the final draft.
- Nicholas Rolnick conceived and designed the experiments, performed the experiments, analyzed the data, prepared figures and/or tables, authored or reviewed drafts of the article, and approved the final draft.
- Brad J. Schoenfeld conceived and designed the experiments, analyzed the data, authored or reviewed drafts of the article, and approved the final draft.
- Ingrid Martins de França conceived and designed the experiments, authored or reviewed drafts of the article, and approved the final draft.
- João Guilherme Vieira conceived and designed the experiments, authored or reviewed drafts of the article, and approved the final draft.
- Amanda Veiga Sardeli conceived and designed the experiments, authored or reviewed drafts of the article, and approved the final draft.
- Okan Kamis conceived and designed the experiments, performed the experiments, authored or reviewed drafts of the article, and approved the final draft.
- Gabriel Rodrigues Neto conceived and designed the experiments, authored or reviewed drafts of the article, and approved the final draft.
- Breno Guilherme de Araújo Tinôco Cabral conceived and designed the experiments, authored or reviewed drafts of the article, and approved the final draft.
- Paulo Moreira Silva Dantas conceived and designed the experiments, authored or reviewed drafts of the article, and approved the final draft.

## Data Availability

The raw data used in our analyses are available in the Supplementary File.

## Supplemental Information

Supplemental information for this article can be found online at http://dx.doi.org/10.7717/peerj.17195#supplemental-information.

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
