# Peer review of "Hypertrophic effects of low-load blood flow restriction training with different repetition schemes: a systematic review and meta-analysis"

_PeerJ, doi:10.7717/peerj.17195_

## Round 0.1 · original submission · Major Revisions

Since reviewer 1 has mentioned a potential issue with duplicate inclusion of a single study, I encourage the authors to check and re-calculate the quantitative part of their analysis. Regarding the methodological approach, did the authors included both between-subject and within-subject trials in the same meta-analysis? This should be avoided given the differing variances within the studies and thus the potential bias on a meta-analysis. Further, it seems that the authors included several values (e.g, locations) from single studies. Since this also has a tremendous effect on the weight of this study within your quantitative approach, I suggest to included a priorization list (also see Gronfeldt et al 2020, or Centner et al. 2020).

·

Basic reporting

No comment.

Experimental design

No comment.

Validity of the findings

No comment at this moment, but see below.

Additional comments

General comments.

After looking through this manuscript and the reference list, I came across a certain case of duplicate publications, i.e., two papers reporting the exact same data. The duplicate publications are the studies of Laurentino et al 2012 & 2022, which both have been included in this meta-analysis.

I offer the following evidence that the aforementioned studies are indeed duplicate publications.

The participant data from Laurentino et al 2012:

Twenty-nine physically active male college students volunteered to participate in this study. Participants from each quartile were then randomly allocated into three groups: low-intensity resistance training (LI) (n = 10, age = 20.3 ± 4.2 yr, weight = 75.3 ± 15.4 kg, height = 175.7 ± 4.9 cm), LI combined with moderate blood flow restriction (LIR) (n = 10, age = 20.0 ± 4.5 yr, weight = 72.1 ± 11.9 kg, height =
175.2 ± 9.0 cm), and high-intensity resistance exercise (HI) (n = 9, age = 23.6 ± 6 yr, weight = 73.8 ± 12 kg, height = 173.6 ± 6 cm).

The participant data from Laurentino et al 2022:

Twenty-nine physically active male college students were randomly allocated into three groups: low-load resistance exercise with blood flow restriction (LL-BFR, n = 10, age = 20.0 ± 4.5 yr, body mass = 72.1 ± 11.9 kg, body height = 175.2 ± 9.0 cm), low-load resistance exercise (LL, n = 10, age = 20.3 ± 4.2 yr, body mass = 75.3 ± 15.4 kg, body height = 175.7 ± 4.9 cm), and high-load resistance exercise (HL, n = 9, age = 23.6 ± 6 yr, body mass = 73.8 ± 12 kg, body height = 173.6 ± 6 cm).

The muscle CSA and strength results from Laurentino et al 2012:

Quadriceps CSA
Both the LIR and the HI groups significantly increased CSA from pre- to posttest (6.3%, P = 0.0007 and 6.1%, P = 0.0004, respectively). No significant difference was observed in CSA from pre- to posttest in the LI group (2.0%, P = 0.9653). However, no significant differences in quadriceps CSA values were detected between groups at posttest (Fig. 1B).

Knee Extension 1RM
All of the groups showed significantly greater 1RM values in the posttest (LI = 20.7%, LIR = 40.1%, and HI = 36.2%) when compared with baseline (main effect for time, P < 0.001). However, no significant differences were detected between groups in the posttest (Fig. 1A). Importantly, the LI group demonstrated a significantly lower delta change in 1RM values (20.7%) when compared with both the LIR and HI groups (36.2% and 40.1%, P = 0.04 and P = 0.0078, respectively).

The muscle CSA and strength results from Laurentino et al 2022:

As expected, 8 weeks of training resulted in significant gains in the muscle CSA (LL pre: 8188 ± 1358, post: 8265 ± 1173, p = 0.965; LL-BFR pre: 7720 ± 1298, post: 8182 ± 1230, p < .0001; HL pre: 7572 ± 1832, post: 8030 ± 1983 mm2, p < .0001) and 1RM (LL pre: 86.2 ± 9.5, post: 106.7 ± 10.8, p < .0001; LL-BFR pre: 84.7 ± 14.5, post: 118.7 ± 16.4, p < .0001; HL pre: 86.9 ± 15.2, post: 118.3 ± 18.9 kg, p < .0001). Moreover, increases in muscle mass (~6.3% vs. 6.1%) and strength (40% vs. 36%) were similar in the LL-BFR and HL groups, respectively, whereas LL resulted in much smaller strength increments (~23%) with no changes in the CSA. However, there was no significant difference between groups in the muscle CSA and 1RM (p > 0.05).

As you can see, both the participant data and the muscle CSA & strength results data are perfect matches. There can be no doubt whatsoever that this is a case of duplicate publications, and yet it also seems clear that Laurentino et al (2022) pretended that their 2022 paper was a new study. The quote below reveals that they discussed their 2012 study as it was a previous study from a different project. This is very unethical practice and arguably a case of scientific misconduct.

"Nonetheless, it has been previously shown that the LL protocol is an insufficient stimulus to induce changes in muscle size or strength when compared with LL-BFR and HL protocols (Laurentino et al., 2012; Takarada et al., 2000)."

The inclusion of duplicate publications is, needless to say, very inappropriate for a meta-analysis and at least one of these studies must therefore be removed from this meta-analysis. This will of course affect the results of this meta-analysis, and I am afraid that the authors will have to do the analyses again. But this is only fair, as the authors should have done a better job in spotting duplicate publications.

Because this affects the entire manuscript, I will refrain from making any other comments on the manuscript this time around and save those for a resubmission of a reworked manuscript.

Reviewer 2 ·

Basic reporting

The article is clear structured in perfect language

Experimental design

Research questions are clearly defined and highly important, especially for clinical applications.

Validity of the findings

Methods and statistical analyses are well conceived and comprehensible

Additional comments

Only minor revisions are required, as noted below:
line 57 please provide a training intensity range for LL-RT?
lines 87-90 Is there any empirical evidence that the the drop out rate is increased in conditions
where BFR is used to failure?
line 166 please provide more specific information about the statistical adjustments
lines 172-173 any good reasons for this?
lines 175-176 revise for clarity, what do you mean with “repeat number categories“? Please use
consisting wording.
line 179 please provide ranges for the practical relevance of effect sizes; when does an effect become
clinically important?
lines 260-261 missing word and letter
lines 352-356 unclear statement, please revise for more clarity

---

## Round 0.2 · Major Revisions

Dear authors, after the withdrawal of reviewer 1, we invited a third reviewer to review your manuscript.

Although reviewer 2 has no additional comments on the manuscript, several major concerns regarding the methodological approach have been raised by reviewer 3.

Reviewer 2 ·

Basic reporting

Dear Editor,
my comments have been sufficiently addressed. In my opinion, this article can be published.Thank you for the opportunity to review this paper.

Experimental design

no new comments

Validity of the findings

no new comments

Additional comments

no new comments

·

Basic reporting

I have no comments regarding this field. I think the manuscript is very well written with clear and concise English. The article is structured according to PRISMA and structured in a professional manner. The references, however, needs to be standardized as fx. some report DOI and some dont

Experimental design

The question is well defined and an update on muscle hypertrophy following BFR compared to HL-RT. The methodology that has been utilized is of high scientific standard. Search terms used in the systematic search is relevant. However, the difference in search terms for Cochrane central is a bit confusing when compared to the other search terms used in the other databases (also mentioned in the attached document). Otherwise, the methods are described in sufficient detail.

Validity of the findings

Although meta-analyses have been performed regarding this question previously, the rationale for this review is warranted. However, regarding the analysis, it seems that the number of participants in the control group in have been extrapolated to entail 9 participants throughout the analysis when compared to different BFR groups in Lixandrao et al. This is adviced against in the Cochrane Handbook. For further detail, see the attached document.

Additional comments

In generel, I think the manuscript is very well written and a good rationale. I think, when the analysis have been corrected, it checks the requirements for publication but will need another round of review.

---

## Round 0.3 · accepted · Accept

I compliment the authors on a well-improved manuscript.

·

Basic reporting

No further comments

Experimental design

No further comments

Validity of the findings

No further comments

Additional comments

No further comments